# Current estimates of $K_1^*$ and $K_2^*$ appear inconsistent with measured $CO_2$ system parameters in cold oceanic regions

Olivier Sulpis[1,2], Siv K. Lauvset[3], Mathilde Hagens[4]

[1]Department of Earth Sciences, Utrecht University, Utrecht, The Netherlands
[2]Department of Earth and Planetary Sciences, McGill University, Montreal, Canada
[3]NORCE Norwegian Research Centre, Bjerknes Centre for Climate Research, Bergen, Norway
[4]Soil Chemistry and Chemical Soil Quality, Wageningen University and Research, Wageningen, The Netherlands

*Correspondence to*: Olivier Sulpis (o.j.t.sulpis@uu.nl)

**Abstract.** Seawater absorption of anthropogenic atmospheric carbon dioxide ($CO_2$) has led to a range of changes in carbonate
chemistry, collectively referred to as ocean acidification. Stoichiometric dissociation constants used to convert measured carbonate system variables (pH, $p$CO$_2$, dissolved inorganic carbon, total alkalinity) into globally comparable parameters are crucial for accurately quantifying these changes. The temperature and salinity coefficients of these constants have generally been experimentally derived under controlled laboratory conditions. Here, we use field measurements of carbonate system variables taken from the Global Ocean Data Analysis Project version 2 and the Surface Ocean $CO_2$ Atlas data products to
evaluate the temperature dependence of the carbonic acid stoichiometric dissociation constants. By applying a novel iterative procedure to a large dataset of 948 surface-water, quality-controlled samples where four carbonate system variables were independently measured, we show that the set of equations published by Lueker et al. (2000), currently preferred by the ocean acidification community, overestimates the stoichiometric dissociation constants at low temperatures, below ~ 8 °C. We apply these newly derived temperature coefficients to high-latitude Argo float and cruise data to quantify the effects on surface-water
$p$CO$_2$ and calcite saturation states. These findings highlight the critical implications of uncertainty in stoichiometric dissociation constants for future projections of ocean acidification in polar regions, and the need to improve knowledge of what causes the $CO_2$ system inconsistencies in cold waters.

# 1 Introduction

In the last decades, oceans have absorbed over a quarter of the anthropogenic carbon dioxide ($CO_2$) emitted to the atmosphere (Le Quéré et al., 2018; Gruber et al., 2019). Upon dissolution in seawater, this $CO_2$ triggers a suite of reactions that lead to a range of chemical changes jointly termed ocean acidification (Zeebe and Wolf-Gladrow, 2001; Gattuso and Hansson, 2011). To accurately calculate the magnitude of these changes, it is crucial to understand the chemical behaviour of $CO_2$ in seawater.

Upon dissolution, $CO_2$ takes the form of solvated $CO_2$ ($CO_{2\ (aq)}$, $CO_2 \cdot H_2O$) or carbonic acid ($H_2CO_3$), which are here both represented by $H_2CO_3^*$, since they can only be readily distinguished by infra-red spectrometry (Zeebe and Wolf-Gladrow, 2001), and the following series of reactions occurs:

$$CO_{2\ (aq)} + H_2O \leftrightarrow H_2CO_3^* \tag{1},$$

$$H_2CO_3^* \leftrightarrow H^+ + HCO_3^- \tag{2},$$

$$HCO_3^- \leftrightarrow H^+ + CO_3^{2-} \tag{3}.$$

Together, these three reactions and their species constitute the marine $CO_2$-$H_2O$ system, which is responsible for about 95% of the acid-base buffering capacity of seawater and maintains the pH of the ocean within a narrow range (Bates, 2019; Zeebe and Wolf-Gladrow, 2001).

Each of these reversible reactions is associated with a thermodynamic equilibrium constant, a number that expresses the relationship between the activities of products and reactants present at equilibrium at a given temperature and pressure. Eqs. (2) and (3) describe the first and second step in the dissociation of carbonic acid. Their equilibrium constants are therefore termed the first and second dissociation constants, $K_1$ and $K_2$, respectively. To avoid the use of activity coefficients, which are not straightforward to derive in seawater, marine scientists have developed a set of *stoichiometric* (or apparent) equilibrium constants to represent the state of the system at a given pressure ($P$), temperature ($T$) and salinity ($S$). To describe the carbonate system, two stoichiometric constants ($K_1^*$ and $K_2^*$, conventionally denoted by a star) are defined in terms of the concentrations of the different species:

$$K_1^* = \frac{[HCO_3^-][H^+]}{[H_2CO_3^*]} \tag{4},$$

$$K_2^* = \frac{[H^+][CO_3^{2-}]}{[HCO_3^-]} \tag{5}.$$

Using these stoichiometric equilibrium constants, we can calculate the relative quantities of the dissolved inorganic carbon ($DIC = [H_2CO_3^*] + [HCO_3^-] + [CO_3^{2-}]$) species. With improved analytical techniques, measurement accuracy of carbonate system variables has substantially increased over the past decades. As a result, uncertainty in carbonate system calculations is currently dominated by the uncertainty of $K_1^*$ and $K_2^*$ values (Orr et al., 2018), justifying the need to investigate whether these uncertainties can be reduced.

Using popular software for carbonate system calculations, e.g., CO2SYS (Lewis and Wallace, 1998; Pierrot et al., 2006; van Heuven et al., 2011) or seacarb (Gattuso et al., 2019), and recently published literature as references, roughly 15 different expressions for $K_1^*$ and $K_2^*$ are currently in use, some of which are (partly) based on refitting data from earlier experiments (e.g. Dickson and Millero, 1987; Lueker et al., 2000); see Table S1 for an overview. Some expressions are based on measurements in artificial seawater of various compositions, while others were carried out in natural seawater. The vast majority of expressions were obtained in the laboratory under controlled conditions, using electrochemical cells either with (e.g. Millero et al, 2006; Millero, 2010) or without (e.g. Roy et al, 1993; Tishchenko et al., 2013) liquid junction. Within these cells, electromotive force readings of equilibrated seawater are used to compute equilibrium constants. Each expression is valid over its own range of $T$ and $S$.

The various expressions for $K_1^*$ and $K_2^*$ obtained this way generally agree well, but discrepancies at low salinities have been highlighted (Cai and Wang, 1998; Millero, 2010; Dinauer and Mucci, 2017; Orr et al., 2018). In addition, the temperature range covered by various $K_1^*$ and $K_2^*$ expressions, although generally broad, only extends below 0 °C in a few studies (Millero et al., 2002; Goyet and Poisson, 1989; Papadimitriou et al, 2018). In fact, Mehrbach et al. (1973), who provided experimental data used by several authors to derive expressions for $K_1^*$ and $K_2^*$ (e.g., Dickson and Millero, 1987; Lueker et al., 2000), used data obtained at only four different temperatures (2, 13, 25 and 35 °C), which brings into question the accuracy of the temperature dependency of these constants. Bailey et al. (2018) recently suggested that the same bias exists for the dissolution of $CO_2$ in seawater and showed that previous expressions of Henry's Law constant for $CO_2$ underestimate the $CO_2$ solubility below 0 °C due to a lack of samples in cold waters. As explained by Raimondi et al. (2019), because the only carbonate system variables currently measured by in situ sensor technologies are $pH$ and the partial pressure of $CO_2$ in seawater ($pCO_2$), relating laboratory or on-board measurements that are usually performed at temperatures ~25 °C to these in situ measurements requires an accurate knowledge of the $K_1^*$ and $K_2^*$ temperature dependency. About 40% of the ocean volume is at an average temperature lower than 2 °C, outside of the temperature range for which the Mehrbach et al. (1973) and derived constants are valid (from the data of Lauvset et al., 2016). An example of this are high-latitude cold waters, which are a critical component of the current global oceanic carbon cycle, as the Southern Ocean surface waters account for ~40% of the annual anthropogenic $CO_2$ uptake by the ocean (Landschützer et al., 2015). Given past difficulties to obtain direct $pCO_2$ measurements from ships in the Southern Ocean (Bakker et al., 2016), a number of autonomous floats have been deployed in the recent years (see, e.g., Williams et al. (2017), Takeshita et al. (2018)). Since these floats estimate $pCO_2$ from a $pH$ measurement and a calculated total alkalinity ($TA$), our knowledge of surface $pCO_2$ in the Southern Ocean strongly relies on the accuracy of dissociation constants in these cold waters.

Best practices for oceanic carbonate system measurements generally recommend the Lueker et al. (2000) constants (Dickson et al., 2007), but the choice for a set of constants may depend on the environment and/or measured carbonate system

variables. Only two of the measurable variables are required to characterize the whole carbonate system, except under conditions where substantial impact of dissolved organic carbon on *TA* is expected (i.e. significant organic alkalinity),. Overdetermination of the carbonate system, i.e., the concomitant measurement of at least three of the carbonate system variables (1) $pCO_2$, (2) *DIC*, (3) *TA* and (4) *pH*, is often used as a tool to identify the best pair of input variables for carbonate system calculations under specific environmental conditions, e.g. in sea-ice brines (Brown et al., 2014) or in systems with substantial organic alkalinity (Koeve and Oschlies, 2012). We refer the reader to Raimondi et al. (2019) for an overview of internal consistency studies, i.e., the agreement between measured and calculated variables. Disagreement between measured and computed values may arise from uncertainties in measurements and, more importantly, equilibrium constants (Orr et al., 2018), but can also result from the choice of relationship between total boron and salinity, as well as organic alkalinity (Fong and Dickson, 2019).

Field measurements are rarely used to derive stoichiometric equilibrium constants because of their interdependence. For example, ship-based measurements of *pH* are normally conducted at fixed temperature (commonly 25°C) and converted to in situ temperature using a second input parameter as well as a set of stoichiometric equilibrium constants (Hunter, 1998). Similarly, using measured *TA* to calculate the contribution of the carbonate system to total alkalinity (carbonate alkalinity, *CA*) requires that the proton concentration and thus *pH* be known (Dickson et al., 2007). To the best of our knowledge, only two studies have so far used overdeterminations of the carbonate system to derive expressions for $K_1^*$ and $K_2^*$ (Millero et al., 2002; Papadimitriou et al, 2018). Both studies used concurrent measurements of $pCO_2$, *TA*, *DIC* and *pH* over a range of temperatures and salinities to calculate $K_1^*$ and $K_2^*$. Millero et al. (2002) used over 6000 sets of pressure-corrected field measurements. They argued that determinations of stoichiometric dissociation constants measured in natural seawater are preferable over those determined in artificial seawater and concluded that the value of $K_2^*$ depends on $pCO_2$, possibly linked to organic alkalinity, which is not accounted for in carbonate system calculations. Papadimitriou et al. (2018), who focussed especially on highly saline brines down to their freezing points, used the same methods as Millero et al. (2002) for their calculations. However, instead of using field measurements, they overdetermined their system under controlled laboratory temperatures and salinities, thus avoiding temperature corrections of the *pH* measurements. Their work, like Orr et al. (2018), confirmed the high uncertainties associated with extrapolating expressions for $K_1^*$ and $K_2^*$ beyond the investigated salinity and temperature ranges.

In the present study, we use the Global Ocean Data Analysis Project version 2 (GLODAPv2, Key et al., 2015; Olsen et al., 2016) and the Surface Ocean $CO_2$ Atlas (SOCAT, Bakker et al., 2016) global data products to constrain stoichiometric equilibrium constants based on surface-water field measurements. Using an iterative procedure that takes into account the lack of independence of *CA* and *pH*, we quantify the temperature dependence of the stoichiometric equilibrium constants. We then use these constants to recommend input pairs for $pCO_2$ and $CaCO_3$ saturation state determinations over various temperature ranges and apply them onto a high-latitude data set.

## 2 Materials and Methods

**2.1 Expressions for $K_1^*$ and $K_2^*$ as a function of carbonate system variables**

Aside from recent advances that allow spectrophotometric determinations of $CO_3^{2-}$ concentrations (Byrne and Yao, 2008; Easley et al, 2013; Sharp and Byrne, 2019), the concentrations of $H_2CO_3^*$, $HCO_3^-$ and $CO_3^{2-}$ are normally not directly measured in seawater. Instead, at least two of the four parameters $pCO_2$, $pH$, $DIC$ and $TA$ are measured, and the concentrations of the individual species inferred from them. In practical terms, $TA$ is the sum of all bases that are titratable with a strong acid

to an equivalence point corresponding to the conversion of $HCO_3^-$ to $H_2CO_3^*$. Here it is defined as:

$$TA = CA + BA + PA + SiA + [OH^-] - [H^+] \qquad (6),$$

with $CA = [HCO_3^-] + 2[CO_3^{2-}]$, and where $BA$ is the borate alkalinity ($[B(OH)_4^-]$), $PA$ is the phosphate alkalinity ($[HPO_4^{2-}]$ + $2[PO_4^{3-}] - [H_3PO_4]$), $SiA$ is the silicate alkalinity ($[SiO(OH)_3^-]$), $[OH^-]$ is the hydroxide ion concentration, and $[H^+]$ is the hydrogen ion concentration. Eq. (6) approximates the definition of $TA$ provided by Dickson (1981), but does not take into account the hydrogen sulphide and ammonia acid-base systems. The terms $BA$, $PA$ and $SiA$ can all be expressed in terms of

stoichiometric equilibrium constants, total concentrations, and $[H^+]$. Hence, knowing $TA$, the total concentrations of dissolved silicate ($[DSi]$), soluble reactive phosphate ($[SRP]$) and boron, as well as $[OH^-]$ and $[H^+]$, $CA$ can be calculated.

To estimate $K_1^*$ and $K_2^*$ as a function of salinity and temperature based solely on independent measurements, we first need to define expressions that define both constants as functions of $CA$, $DIC$, $pCO_2$ and $[H^+]$. Both $K_1^*$ and $K_2^*$ are normally

defined in terms of proton concentration, $[H^+]$, and the acid-base species they describe, see Eq. (4) and (5). In this work, we replace $[H_2CO_3^*]$, $[HCO_3^-]$ and $[CO_3^{2-}]$ by expressions that only contain the four variables present in the dataset and Henry's constant, $K_0$, taken from Weiss (1974). This leads to the following set of equations, which are equivalent to those presented in Millero et al. (2002) and Papadimitriou et al. (2018):

$$K_1^* = \frac{[H^+](2DIC - CA - 2K_0 pCO_2)}{K_0 pCO_2} \qquad (7),$$

$$K_2^* = \frac{[H^+](CA - DIC + K_0 pCO_2)}{2DIC - CA - 2K_0 pCO_2} \qquad (8).$$

Note that similar expressions can also be derived when only three independently measured variables are available in the dataset. In this case, either $K_1^*$ or $K_2^*$ remains in the expression, in addition to any three variables of the set $CA$, $DIC$, $pCO_2$ and $[H^+]$. Derivations of all these expressions can be found in the supplementary information.

**2.2 Data**

Data for $T$, $P$, practical salinity ($S_P$), $DIC$, $TA$, $pH$, $[SRP]$ and $[DSi]$ were taken from GLODAPv2 (Key et al., 2015; Olsen et al., 2016). Only data associated with a WOCE flag of 2 were retained for this analysis. WOCE (World Ocean

Circulation Experiment) flags are associated to each GLODAPv2 variable during quality control. Data associated with a flag of 2 were assessed as "acceptable" by quality controllers of the original dataset. Re-calculated or estimated variables and
samples with missing $T$ or $S_P$ were always discarded. In total, we obtained 98326 samples for which $TA$, $DIC$ and $pH$ are available from independent, high-quality measurements.

         $pCO_2$ values were obtained from SOCAT version 3 (Bakker et al., 2016). Only data associated with a WOCE flag of 2 are used. When available, a $pCO_2$ value was selected and added to corresponding surface GLODAPv2 samples. To select
the most accurate $pCO_2$ value, we only merged GLODAPv2 and SOCAT samples from the same cruise and taken within the same hour; in most cases within the same 20 min. As a result, we assembled 1024 samples for which $TA$, $DIC$, $pH$ and $pCO_2$ are all available from independent high-quality measurements. As underway $pCO_2$ measurements available in the SOCAT database are all from the surface ocean, it was not possible to assign measured $pCO_2$ values to samples at depth. Note that we discarded data from two cruises (EXPOCODES 33AT20120419 and 49NZ20010828) for reasons explained in the
supplementary information, ultimately using data from 948 samples for this analysis. Samples within this dataset were taken between 1993 and 2012 over 26 different research cruises. These samples were taken at the ocean surface, always in the top 5 meters. They cover a range of practical salinities from 30.73 to 37.57 and temperatures from -1.67 to 31.80 °C, at locations shown in Fig. 1.

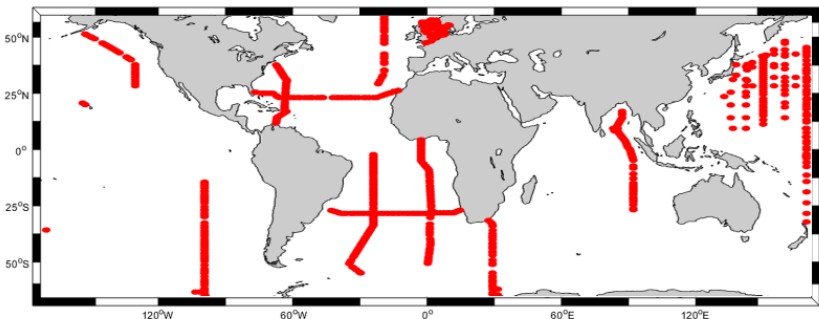


**Figure 1.** *Spatial coverage of the dataset containing GLODAPv2 and SOCAT samples for which DIC, TA, pH and pCO₂ are available from independent, high-quality measurements.*

**2.3 Iterative methods and underlying assumptions**

180        To the 948 samples for which independent, high-quality measurements of $pH$, $DIC$, $TA$, $[DSi]$, $[SRP]$ and $pCO_2$ are available, we would preferably directly apply Eq. (7) and (8). This was, however, not possible given the interdependence of $pH$ and $CA$, both of which are necessary to compute $K_1^*$ and $K_2^*$ and, in turn, other carbonate system parameters. Rather than estimating the temperature dependence of $pH$ from $\Delta pH/\Delta T$ as done by Millero et al. (2002), we used a novel iterative fitting procedure. This procedure is based on an initial estimate of both $pH$ and $CA$ using the Lueker et al. (2000) constants, followed

by a re-computation at each iteration using the values of $K_1^*$ and $K_2^*$ from the previous iteration. The calculations were executed in R (R Core Team, 2019) and detailed below. The code and data files can be downloaded from an online repository (https://doi.org/10.5281/zenodo.3725889).

    Since the objective of the present study is to obtain independent measurements of $K_1^*$ and $K_2^*$, we could not directly

use GLODAPv2 in situ *pH* data because the majority of these data were obtained on board by potentiometric or spectrophotometric methods at an equilibrium temperature often higher than the surface seawater (usually 25°C, occasionally 20 or 13°C). In addition, *pH* data were not always delivered to GLODAPv2 on the total *pH* scale. Consequently, the in situ *pH* measurements available in GLODAPv2 are all recalculated using measured *TA* and the Lueker et al. (2000) stoichiometric dissociation constants, and converted to the total *pH* scale if necessary (Olsen et al., 2016). To obtain the *pH* values delivered

to GLODAPv2, which should be independent of $K_1^*$ and $K_2^*$, we converted the GLODAPv2 *pH* values back to their measurement temperatures (*labT*) and pH scales, as recorded in the cruise reports. For this reconversion, we used the bias-corrected *TA* value from GLODAPv2 rather than the measured *TA* values that were used in the GLODAPv2 conversion process. Bias correction of *TA* was done through crossover and inversion analysis of the data (Olsen et al., 2016); for the 26 research cruises we selected, bias correction resulted in *TA* adjustments of -1 to 10 µmol kg$^{-1}$. These adjustments however

affected the recalculated *pH* values by less than 0.0001. We then converted all recalculated *pH* values to the free *pH* scale ($pH_F^{labT}$) using the default settings of the *pHconv* function in the seacarb R package (Gattuso et al., 2019). The free *pH* scale was used during the fitting procedure to avoid further complications with the sulphate and fluoride acid-base systems. Nevertheless, final results are presented on the total *pH* scale.

Carbonate alkalinity was not directly measured, but is instead a *pH*-dependent quantity computed from *TA*, see Eq. (6). As a first approximation, we calculated *CA* from the measured *TA* by subtracting the contributions of the borate, silicate and phosphate acid-base systems, as well as the auto-dissociation of water, using *[SRP]*, *[DSi]* and the in situ *pH* from GLODAPv2. We estimated the total boron concentration from salinity using the Uppström (1974) relationship and calculated its acid-base speciation using the equilibrium constants of Dickson (1990). For the silicate and phosphate acid-base speciation,

the equilibrium constants of Yao and Millero (1995) were used. All of these expressions are only valid for temperatures above 0°C; thus, extrapolation to lower temperatures yields an additional uncertainty to the method. All equilibrium constants were corrected for pressure following Millero (1995), but given that all samples were taken at depth shallower than 5 m depth, this correction is negligible. Using $H_F^{labT}$, the proton concentration computed from $pH_F^{labT}$, we also calculated carbonate alkalinity at the temperature of *pH* measurements ($CA^{labT}$). This variable was used during the iteration procedure.


    Since there are two *pH*-independent parameters (*DIC*, $pCO_2$), we can use these two parameters and one *pH*-dependent parameter (either *pH* itself, or *CA*) to initialise the iterative procedure. This implies that either $K_1^*$ or $K_2^*$ must be assigned an initial value before starting the iterations. Here, we initially set the in situ $K_2^*$ to the value calculated from the Lueker et al.

(2000) expressions. The alternative case in which in situ $K_1^*$ was initially set to the Lueker et al. (2000) value is described in the supplementary information and would have no appreciable impact on the results presented here, i.e., whether the first or the second dissociation constant is assigned an initial value does not affect the results. Each iteration consisted of four different steps:

(1) First, $K_1^*$ was computed from in situ $DIC$, $CA$, $pCO_2$ and $K_2^*$ from Lueker et al. (2000) using Eq. (7) and a Newton-Raphson technique (function *uniroot.all* from package rootSolve, Soetaert and Herman, 2009). These calculated $K_1^*$ values were subsequently fitted to a general expression as a function of temperature and salinity of the form:

$$pK_{1\ or\ 2}^* = a_1 + a_2 S_P + a_3 S_P^2 + \frac{a_4}{T} + a_5 \ln(T) \tag{9},$$

where $pK_i^*$ corresponds to $-\log_{10}(K_i)$ and $a_i$ are fitting coefficients determined using the Levenberg-Marquardt algorithm for nonlinear least-squares estimates (function *nlsLM* from the minpack.lm package, Elzhov et al., 2016). This expression is of a similar form as Lueker et al. (2000), to facilitate the comparison. Because the salinity range in the sub-dataset where four carbonate system variables are available is narrow (30.73 to 37.57), it was not possible to obtain converging iterations where all the coefficients in Eq. (9) were resolved. Thus, we kept $a_2$ and $a_3$ fixed to the Lueker et al. (2000) values, assuming that the salinity dependence of $K_1^*$ and $K_2^*$ is correct for the salinity range of our dataset, and only solved for $a_1$, $a_4$ and $a_5$.

(2) Second, this new expression for $K_1^*$, as well as $CA$ and the expression for $K_2^*$ used in step 1, were used to compute $pH$ at in situ temperature. For this, both $K_1^*$ and $K_2^*$ were calculated at the temperature of $pH$ measurement ($K_1^{*,labT}$ and $K_2^{*,labT}$). These were used together with the free proton concentration at lab temperature ($H_F^{labT}$) and the calculated carbonate alkalinity ($CA^{labT}$), both of which do not change during the iterative procedure, to calculate $[H^+]$, the free proton concentration at in situ temperature. We expressed $DIC$ as a function of $CA$, $[H^+]$, $K_1^*$ and $K_2^*$, and assumed that the value of $DIC$ is independent of temperature. Thus,

$$CA \frac{[H^+]^2 + [H^+]K_1^* + K_1^* K_2^*}{[H^+]K_1^* + 2K_1^* K_2^*} = CA^{labT} \frac{(H_F^{labT})^2 + H_F^{labT} K_1^{*,labT} + K_1^{*,labT} K_2^{*,labT}}{H_F^{labT} K_1^{*,labT} + 2K_1^{*,labT} K_2^{*,labT}} \tag{10}$$

This equation was rewritten into a quadratic equation, solved analytically for $[H^+]$, and converted to $pH$.

(3) Third, $CA$ – which is dependent on $pH$ – was updated based on the new $[H^+]$, as per Eq. (6) and the method outlined for the initial calculation of $CA$.

(4) Fourth, we used Eq. (8) to calculate $K_2^*$ as a function of $pCO_2$, $DIC$, the new $pH$ and $CA$, and fit these in situ computed constants to an equation of the form of Eq. (9).

These four steps were repeated and at each iteration, $K_2^*$, $CA$, and $pH$ from the previous iteration, were used as initial values. Note that this method assumes that the uncertainty in $K_0$ is minor compared to that in $K_1^*$ and $K_2^*$. We also assumed that no acid-base systems other than the carbonate, borate, silicate and phosphate acid-base systems contributed to $TA$ – this point will be elucidated later – and that uncertainties in the calculated contributions of the latter three acid-base systems to $TA$ were also minor compared to the uncertainties in $K_1^*$ and $K_2^*$.

**2.4 Uncertainty propagation**

The overall uncertainty on the final $K_1^*$ and $K_2^*$ values is a combination of the uncertainties associated with measurement errors (hereafter termed "analytical uncertainty") and the uncertainties resulting from the fitting procedures (hereafter termed "fitting uncertainty"), that are propagated throughout the iterations. The analytical uncertainty ($\sigma K^{ana.}$) was computed using the predefined accuracy limits (here, for simplicity, denoted $\sigma$) used for the GLODAPv2 secondary quality
control procedures. This accuracy limit reflects the minimum bias that can be detected with reasonable certainty (Tanhua et al., 2010) and is based on an objective analysis of systematic biases in ship-based data. Within the GLODAP context the accuracy limit should be interpreted as "the range within which we can realistically expect measurements from the deep ocean to be reproducible". For each variable the corresponding value is taken from Table 2 in Olsen et al. (2016), i.e., $\sigma S_P = 0.005$, $\sigma[DSi] = 2\%$, $\sigma[SRP] = 2\%$, $\sigma DIC = 4$ µmol kg$^{-1}$, $\sigma TA = 6$ µmol kg$^{-1}$. $\sigma pH$ is set to 0.01 following Table 3 in Olsen et al.
(2019) and $\sigma pCO_2$ is set to 2 µatm, corresponding to the minimum accuracy of SOCAT quality control flags A or B. While referred to as accuracy this number is actually a measure of overall measurement uncertainty, and includes uncertainties due to environmental factors (Pierrot et al., 2009). $\sigma CA$ was computed as the square root of the sum of the squares of $\sigma CO_3^{2-}$ and $\sigma HCO_3^-$. In turn, these were computed using $TA$, $pH$, $[SRP]$, $[DSi]$, $P$, $T$ and $S_P$ as input variables, as well as their respective aforementioned uncertainties, using the error propagation code of Orr et al. (2018). The analytical uncertainty on both $K_1^*$ and
$K_2^*$ was then estimated following the standard rules of error propagation, as per the following equations:

$$\sigma K_1^{ana.} = K_1^* \sqrt{\left(\frac{\sigma[H^+]}{[H^+]}\right)^2 + \left(\frac{\sigma fCO_2}{fCO_2}\right)^2 + \left(\frac{\sqrt{(2\sigma DIC)^2 + (\sigma CA)^2 + (2K_0\sigma fCO_2)^2}}{2DIC - CA - 2K_0 fCO_2}\right)^2} \quad (11),$$

$$\sigma K_2^{ana.} = K_2^* \sqrt{\left(\frac{\sigma[H^+]}{[H^+]}\right)^2 + \left(\frac{\sqrt{(\sigma CA)^2 + (\sigma DIC)^2 + (K_0\sigma fCO_2)^2}}{CA - DIC + K_0 fCO_2}\right)^2 + \left(\frac{\sqrt{(2\sigma DIC)^2 + (\sigma CA)^2 + (2K_0\sigma fCO_2)^2}}{2DIC - CA - 2K_0 fCO_2}\right)^2} \quad (12).$$

The fitting uncertainty ($\sigma K^{fit.}$) was obtained using a Monte Carlo simulation technique that propagates errors in the
fitting coefficients to the predicted K values. At the end of the iterations, the non-linear least-square model fits obtained with the *nlsLM* function were used as an input in the *predictNLS* function, from the *propagate* R package (Spiess, 2018), to calculate $\sigma K^{fit.}$, neglecting any error in the temperature measurements. The overall uncertainty on $K_1^*$ and $K_2^*$ was then assumed to be the square root of the sum of the squares of the analytical and fitting uncertainties. The 95% confidence intervals for each of

the fitting coefficients, i.e., $a_i$ in Eq. (9), shown in Table 1, were extracted from the result of the non-linear least-squares model fits in R using the *summary* function. Note that, because we did not solve for the salinity coefficients in Eq. (10) due to the limited salinity range of the four carbonate-system variables dataset, $a_2$ and $a_3$ are set to the Lueker et al. (2000) values and no confidence interval is computed for these coefficients.

## 3 Results

Using all the data from which $T$, $S_P$, $DIC$, $pH$, $CA$ and $pCO_2$ are available as high quality, independent measurements, we were able to derive expressions for $K_1^*$ and $K_2^*$ with a new temperature dependence. The coefficients $a_i$ for $pK_1^*$ and $pK_2^*$, in an equation of the form of Eq. (9), and after 30 iterations, are reported in Table 1, along with their respective 95% confidence intervals. In both expressions, all coefficients are significantly different from zero (p values < 0.001). As $CA$ and $pH$ are being updated throughout the iterations, $CA$, $pH$, $K_1^*$ and $K_2^*$ all evolved until the 10th iteration, before converging to a final value (Fig. 2). Thus, we stopped the iterative process after 30 iterations. Between the initial GLODAPv2 value and the 30th iteration, $[H^+]$ values vary by up to 6.6%. $CA$ values are only weakly affected by these in situ $pH$ updates as they change by a maximum of 0.2% throughout the first 30 iterations. Largest $pH$ and $CA$ changes occur in the colder end of the temperature range. $pK_1^*$ and $pK_2^*$ values both shift upward throughout the iterations, the $pK_1^*$ increase being higher than that of $pK_2^*$, especially in cold waters (Fig. 2).

**Table 1.** *Comparison of the coefficients for $pK_1^*$ and $pK_2^*$ between this study and Lueker et al. (2000), using an equation of the form of Eq. (9). Coefficients are given as value ± 95% confidence interval.*

| | This study | | Lueker et al. (2000) | |
| --- | --- | --- | --- | --- |
| | *pK₁\** | *pK₂\** | *pK₁\** | *pK₂\** |
| $a_1$ | - 172.4493 ± 26.131 | - 59.4636 ± 24.016 | - 61.2172 | 25.9290 |
| $a_2$ | - 0.011555 | - 0.01781 | - 0.011555 | - 0.01781 |
| $a_3$ | 0.0001152 | 0.0001122 | 0.0001152 | 0.0001122 |
| $a_4$ | 8510.63 ± 1139.8 | 4226.23 ± 1050.8 | 3633.86 | 471.78 |
| $a_5$ | 26.32996 ± 3.9161 | 9.60817 ± 3.5966 | 9.67770 | - 3.16967 |

As shown in Fig. 3, the $pK^*$ values obtained with the iterative procedure are statistically indistinguishable from the $pK^*$ values of Lueker et al. (2000) (i.e., pK$^{this\ study}$ - σpK$^{this\ study}$ < pK$^{Lueker}$ < pK$^{this\ study}$ + σpK$^{this\ study}$) over most of the temperature range. Nevertheless, in cold waters, below temperatures 8.1 and 9.2°C respectively, $pK_1^*$ and $pK_2^*$ are significantly higher than the values reported by Lueker et al. (2000). In Fig. 3, we also provide a comparison of the $pK^*$ values of this study with those

of Papadimitriou et al. (2018), as the latter study focuses on low-temperature waters. As shown in Figs. 3c,d, both $pK_1^*$ and $pK_2^*$ from Papadimitriou et al. (2018) are slightly lower than the fitted values from this study, for a similar salinity, and are also lower than the Lueker et al. (2000) $pK^*$ values except for temperatures below 4.0 and 1.2°C, respectively. Thus, that the Lueker et al. (2000) study underestimates both $pK_1^*$ and $pK_2^*$ in waters near freezing point seems to be a consistent feature across studies.

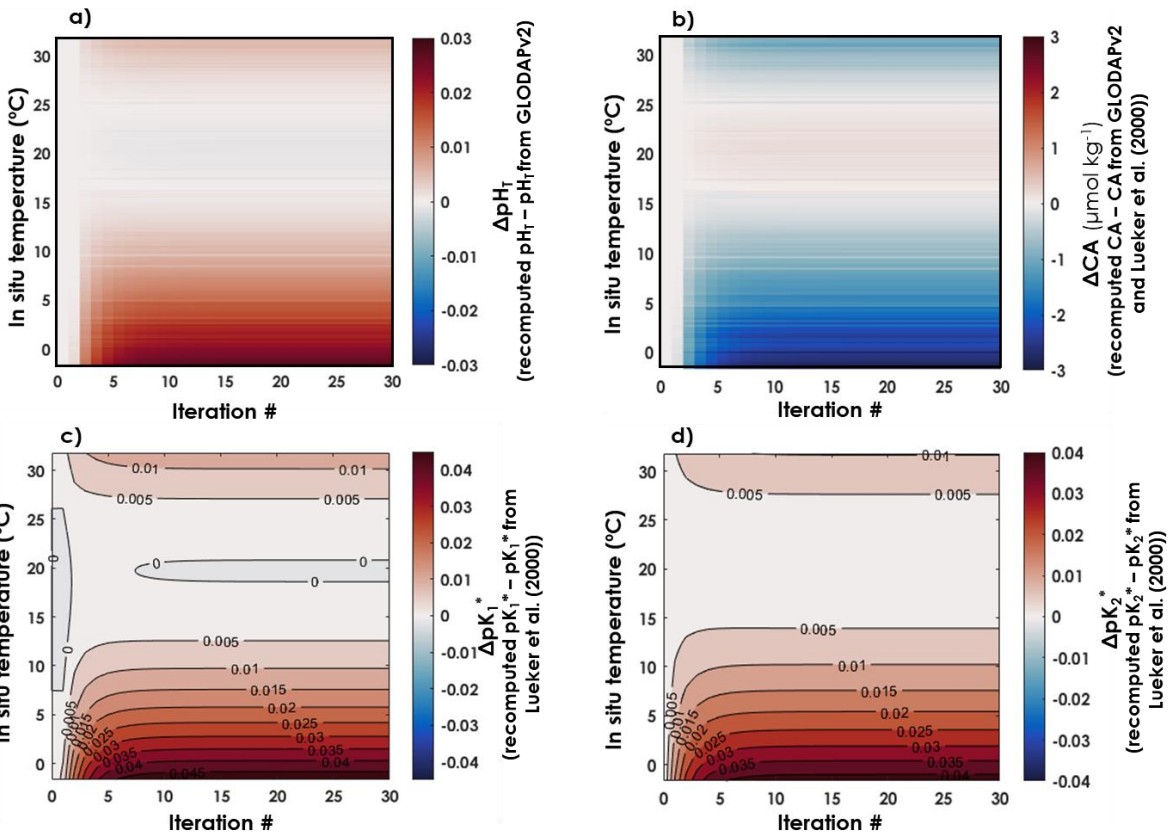

***Figure 2.*** *Evolution of pH$_T$, CA, pK$_1^*$ and pK$_2^*$ as a function of in situ temperature and iterations. a) Differences between recomputed pH$_T$ and pH$_T$ from GLODAPv2. b) Differences between recomputed CA and CA estimated from GLODAPv2 data and the Lueker et al. (2000) constants. Differences between recomputed c) pK$_1^*$ and d) pK$_2^*$ and the Lueker et al. (2000) values at in situ temperature and at a practical salinity of 35.*

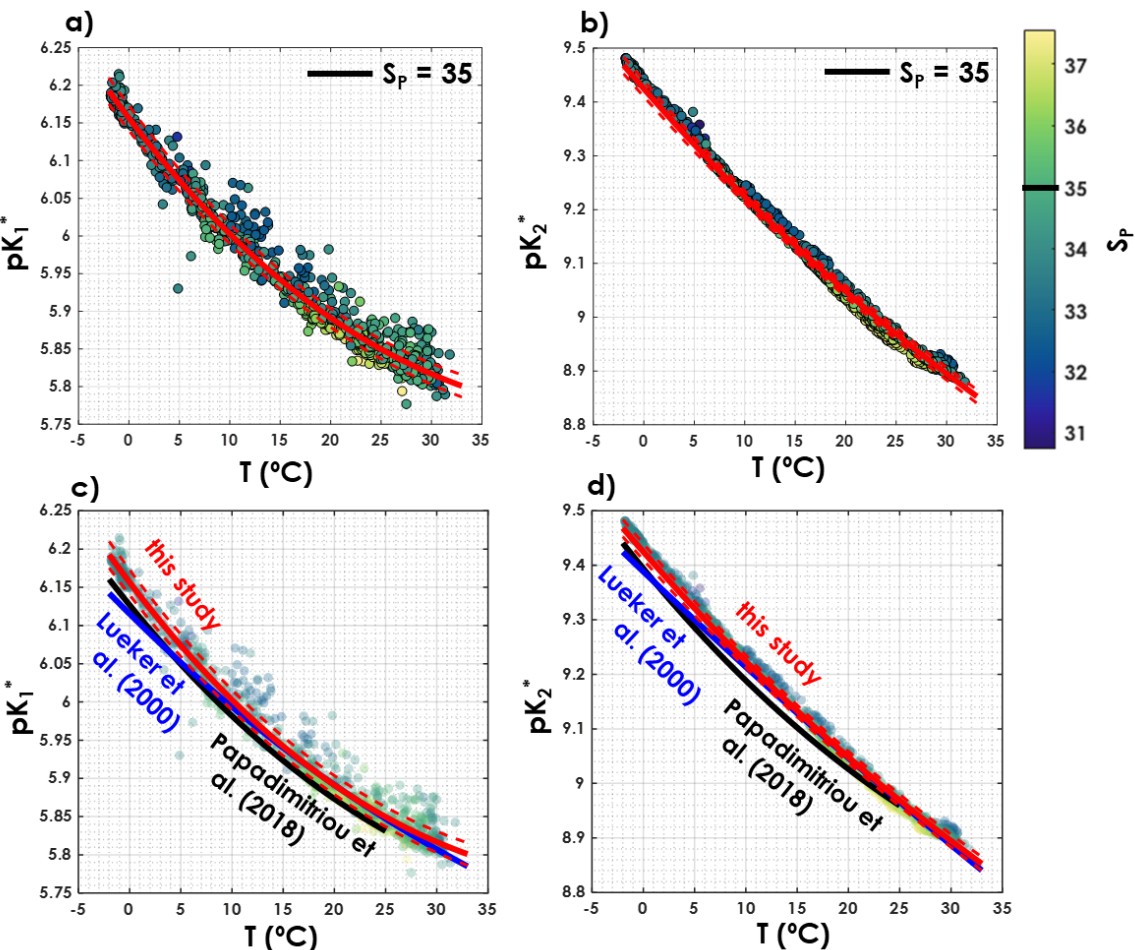

***Figure 3.*** *(**a**) $pK_1^*$ and (**b**) $pK_2^*$ as a function of temperature where the colour represents practical salinity ($S_P$), and the fits are fixed for a $S_P$ of 35. Comparison of (**c**) $pK_1^*$ and (**d**) $pK_2^*$ as a function of temperature from this study (red lines), Lueker et al. (2000, blue line) and Papadimitriou et al. (2018, black line). The solid blue line represents the $pK^*$ fits from Lueker et al. (2000), the solid red line the $pK^*$ from this study computed with the coefficients presented in Table 1. Dashed red lines are overall uncertainties as defined in section 2.4.*

## 4 Discussion

Using underestimated $pK_1^*$ and $pK_2^*$ values implies that, for a given state, computed $[H_2CO_3^*]$ or $pCO_2$ would be underestimated and $[CO_3^{2-}]$ overestimated. This potentially has strong implications for our representation of seawater carbonate chemistry in low-temperature marine environments, such as polar regions. Hence, we highlight the implications of this work for the estimation of two carbonate system variables in polar regions, i.e. $pCO_2$ and the saturation state of seawater with respect to calcite ($\Omega_{Ca}$). But first, we examine error propagation and the dependence of $pK_1^*$ and $pK_2^*$ to salinity, and discuss the influence of organic alkalinity and the quality of pH measurements on the results presented here.

### 4.1 Influence of *pH* and *TA* measurement quality

The *pH* of samples that are used to derive the $K^*$ fits presented here was measured using electrodes or spectrophotometrically, between 1993 and 2012. During this period, it was shown that impurities present in commercially available dyes could generate a systematic bias in the measured *pH* (Yao et al., 2007). Recently, Carter et al. (2018) pointed to systematic discrepancies resulting from differing approaches to *pH* measurements. Yao et al. (2007) noted that these impurities can contribute to *pH* offsets as large as 0.01 *pH* units, which corresponds to the analytical uncertainty in *pH* ($\sigma pH$) that we use here, taken from Olsen et al. (2019). We must therefore investigate whether the fact that most *pH* measurements in the *pH* dataset are not from spectrophotometric measurements with purified dyes could alter the conclusion of underestimated $pK^*$ values in cold waters. To answer this question, we gathered a sub-dataset of the more recent GLODAPv2.2019 data product (Olsen et al., 2019), composed of samples from 9 different cruises (EXPOCODES 320620140320, 06AQ20150817, 33AT20120324, 33AT20120419, 33HQ20150809, 33RO20150410, 33RO20150525, 33RO20161119 and 33RR20160208) for which *T*, *P*, $S_P$, *DIC*, *TA*, *[SRP]* and *[DSi]* are available and associated with a WOCE flag of 2, for which *pH* was measured spectrophotometrically using purified dyes only, and for which an associated SOCAT $pCO_2$ value is available. Although this independent dataset is too small to apply the iterative procedure and obtain acceptable $pK^*$ fits, we can use it to compare the $K_1^*$ / $K_2^*$ values obtained with Eqs. (7, 8) and this purified-dye independent dataset to the $K_1^*$ / $K_2^*$ values obtained with Eqs. (7, 8) and the regular dataset.

For both the purified-dye independent dataset and the regular dataset, samples were sorted according to in situ temperature and grouped into bins of 0.5 °C. Temperature bins containing a single sample were not used. For each temperature bin, the mean $pK^*$ values, obtained with Eqs. (7,8), along with the associated uncertainties, were computed. Plotting the difference between the $pK^*$ values computed using the regular dataset and those computed using the purified-dye independent dataset as a function of seawater in situ temperature (Fig. 4), we do not see any clear systematic bias caused by the use of purified dyes. This means that $pK^*$ values computed from a dataset with purified-dye pH measurements only are not higher or lower than $pK^*$ values computed from a dataset with pH measured using primarily impure dye. More importantly, in colder waters (T < ~ 2 °C), the differences between the $pK^*$ values from this study and those from Lueker et al. (2000) (black line in

Fig. 4) are larger than what can be explained by the choice of dye for spectrophotometric *pH* measurements. Thus, the use of impure vs. purified dye in *pH* measurements should not affect the conclusions presented here.

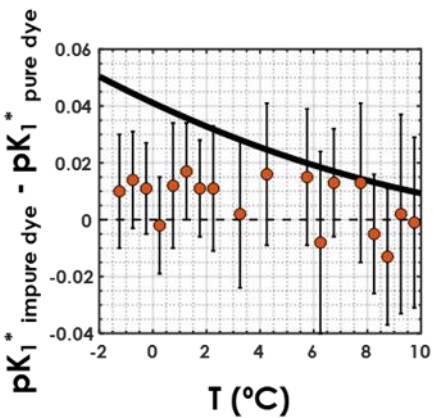 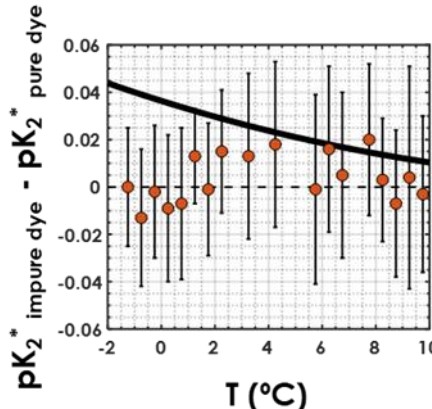


***Figure 4.*** *(left) $pK_1^*$ and (right) $pK_2^*$ differences between those obtained with the regular dataset and those obtained with the purified-dye independent dataset, as a function of in situ seawater temperature. Orange circles represent the mean computed $pK^*$ values within a temperature bin, and the vertical black bars stand for the associated uncertainties deviations. The solid black line is the difference between the $pK^*$ fit from this study and that from Lueker et al. (2000).*


Another issue in GLODAPv2 carbonate system measurements may be the fact that some seawater samples contain measurable amounts of organic bases (Fong and Dickson, 2019; Patsavas et al., 2015; Yang et al., 2015). This organic alkalinity
is unaccounted for in the definition of total alkalinity of Eq. (6), thus causing biased, overestimated, computed carbonate alkalinity values. This does not only concern coastal waters, but also open-ocean waters, where the total concentration of these organic bases could be in the order of a few µmol kg$^{-1}$ (Fong and Dickson, 2019). This, however, should not substantially alter the results presented here, due to the small amount of these organic bases and consequently small impact on computed $pK^*$ values. If any, subtracting the contributions of these unaccounted bases to the total alkalinity measurements would have a
unidirectional effect on the dissociation constant estimates, shifting the $pK^*$ values upwards – see Eqs. (7,8) – further away from the Lueker et al. (2000) values in cold waters.

## 4.2 Uncertainties in carbonate system calculations

The relative overall uncertainties ($\sigma K^* / K^*$) were ~2.5% for both $K_1^*$ and $K_2^*$. In both cases, the analytical uncertainty
($\sigma K^{ana.}$) was more than twice as high as the fitting uncertainty ($\sigma K^{fit}$). The overall *pH* measurement uncertainty of the GLODAP dataset ($\sigma pH = 0.01$) is relatively high, causing the *pH* term in Eqs. (11,12) to be the dominant factor in $\sigma K^{ana.}$. Thus, what

dominates the overall uncertainties for $K^*$ estimates from this study is the uncertainty in $pH$, which explains the fact that the relative uncertainties of $K_1^*$ and $K_2^*$ are similar. Converting from $\sigma K^*$ to $\sigma pK^*$, we find $\sigma pK_1^* = \sigma pK_2^* = 0.011$. The overall uncertainty for $pK_1^*$ is higher than that reported by Orr et al. (2018), but the overall uncertainty for $pK_2^*$ is smaller. These

values are quite high relative to the uncertainties of previous expressions for $K_1^*$ and $K_2^*$ as reported in Table 2 of Millero (2007), which we attribute to the fact that they reflect both the uncertainty from the fits and from the measurements that we use.

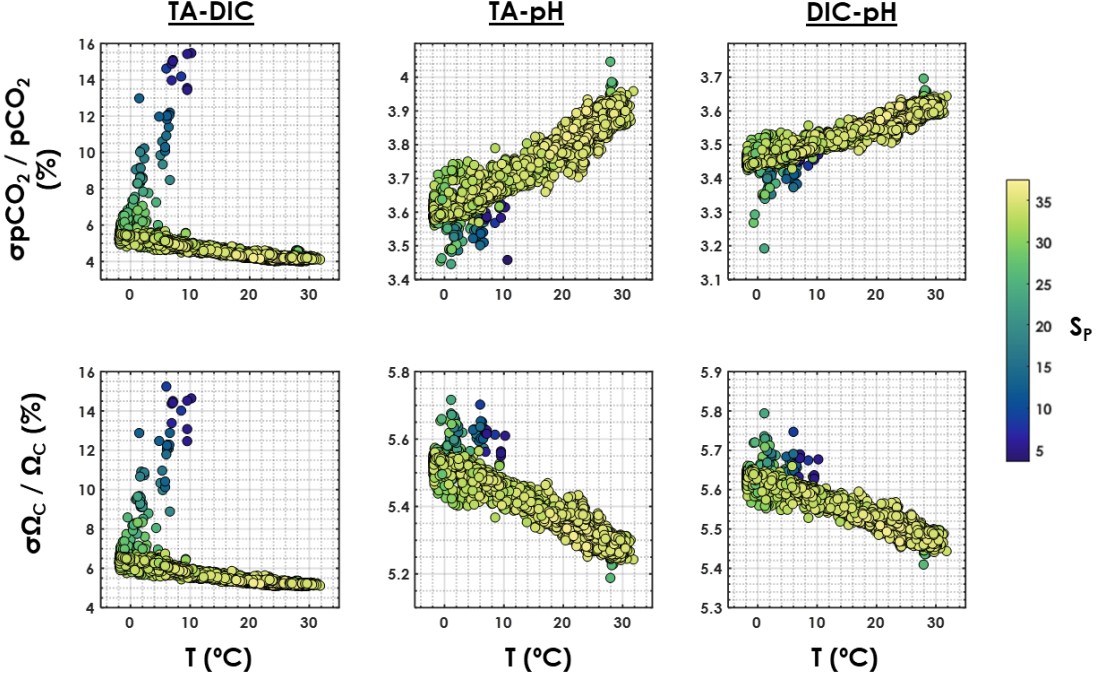

**Figure 5.** *Overall relative uncertainty for (top) pCO₂ and (bottom) Ω$_{Ca}$ as a function of in situ temperature (horizontal axis) and salinity (colour bar). The left column is for the TA-DIC pair, the central column is for the TA-pH pair and the right column is for the DIC-pH pair. This is computed using all data points from GLODAPv2 that contain T, P, S$_P$, DIC, TA, pH, [SRP] and [DSi], in the top 10 meters of the water column (3392 samples).*


Using the $pK^*$ values from Table 1, setting the analytical uncertainties for each variable to the values reported in section 2.4, and using $\sigma pK_1^* = \sigma pK_2^* = 0.011$, we use the Excel version of CO2SYS from Orr et al. (2018) and analyse the propagation of uncertainties on two computed variables, $pCO_2$ and the saturation state of seawater with respect to calcite, $\Omega_{Ca}$ (Mucci et al., 1983, see discussion in section 4.3). For this purpose, we use all data points from GLODAPv2 that contain $T$, $P$,

$S_P$, $DIC$, $TA$, $pH$, $[SRP]$ and $[DSi]$, in the top 10 meters of the water column. The quality criteria remains unchanged, i.e., we use only data associated with a WOCE flag of 2.. The obtained dataset contains 3392 samples, including the 948 samples of

the regular dataset, covers a salinity range from 3.46 to 37.57 and a temperature range from -1.91 to 31.80 °C. Depending on which carbonate-system pair of variables is used, both the magnitude and the uncertainties of computed variables can differ (Orr et al., 2018; Ribas-Ribas et al., 2014). Here, we use three different pairs of variables, i.e. *TA-DIC*, *TA-pH* and *DIC-pH*, to compute $pCO_2$ and $\Omega_{Ca}$ and their associated propagated uncertainties, $\sigma pCO_2$ and $\sigma \Omega_{Ca}$, respectively.

Relative uncertainties generated with the *TA-DIC* pair appear to be particularly sensitive to salinity (Fig. 5), increasing from ~5 to ~15% for both $pCO_2$ and $\Omega_{Ca}$ as the salinity decreases from 35 to 5. The overall relative uncertainty for both $pCO_2$ and $\Omega_{Ca}$ is less dependent on salinity when pH is used as an input variable with either *DIC* or *TA*. For both the *TA-pH* and the *DIC-pH* pairs, the overall relative uncertainty on $pCO_2$ increases with an increasing temperature, while the overall relative uncertainty on $\Omega_{Ca}$ decreases with an increasing temperature (Fig. 5).

As depicted in Fig. 6, in agreement with Raimondi et al. (2019), we conclude that the *DIC-pH* pair offers the lowest overall relative uncertainty for computed $pCO_2$ over the range of salinities and temperatures investigated. Using the *DIC-pH* pair also has the important benefit of not having to make any assumption regarding organic alkalinity or the boron to salinity ratio (Fong and Dickson, 2019). Conversely, the *TA-pH* pair is the one generating the highest overall relative uncertainties on computed $pCO_2$. As for $\Omega_{Ca}$, the *TA-pH* pair provides the lowest overall relative uncertainty below a temperature of ~20 °C, whereas the *TA-DIC* pair should be preferred in warmer waters (Fig. 6).

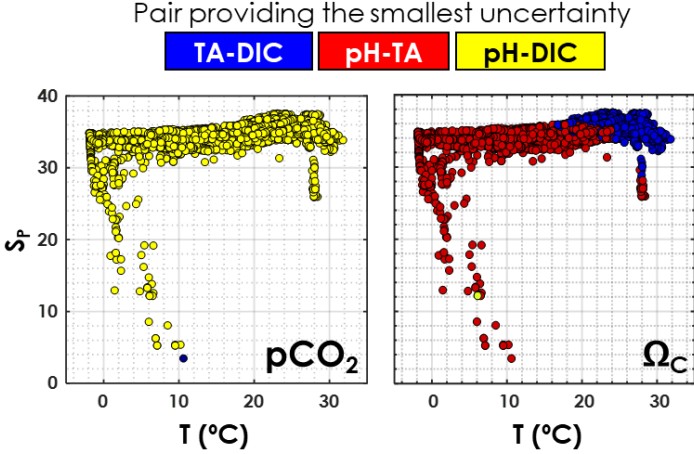

**Figure 6.** *Pair of carbonate system variables (TA-DIC, TA-pH or DIC-pH) providing the lowest overall relative uncertainty, as a function of in situ temperature and practical salinity, for pCO₂ and $\Omega_{Ca}$. This is computed using all data points from GLODAPv2 that contain T, P, S_P, DIC, TA, pH, [SRP] and [DSi] in the top 10 meters of the water column (3392 samples).*

**4.3 Implications for surface ocean $p$CO$_2$**

To evaluate the implications of the revised temperature dependence of the carbonic acid dissociation constants, we compare ocean carbonate chemistry as calculated with the constants from this study, those of Lueker et al. (2000) and those of Millero et al. (2002). Whereas the constants from Lueker et al. (2000) are the most commonly used by the oceanographic community, as recommended by Dickson et al. (2007), the constants from Millero et al. (2002) were derived in an approach similar to that presented here, using a large dataset of in situ measurements. Thus, it appears relevant to include a comparison with Millero et al. (2002) in the present discussion. The major differences between our approach and the approach of Millero et al. (2002) are the calculation of *CA* from measured *TA* and *pH* (iteratively versus direct) and the conversion of pH measurements (iteratively versus estimating $\Delta pH/\Delta T$ from the constants of Mehrbach et al. (1973)).

For this comparison, we use data from the Southern Ocean Carbon and Climate Observations and Modelling (SOCCOM) project (https://soccom.princeton.edu/). The SOCCOM project has deployed more than 100 Argo floats equipped with biogeochemical sensors in the Southern Ocean. These sensors include *pH*, and SOCCOM routinely calculate the full carbon chemistry (including *pCO$_2$* and $\Omega_{Ca}$) using a combination of measured *T*, *S$_P$*, *pH*, *O$_2$*, and empirical algorithms for *TA* (Carter et al., 2018). The SOCCOM data used here, both measured and calculated, were downloaded as a Matlab file from https://library.ucsd.edu/dc/object/bb0515927k.

The method used to calculate *pCO$_2$* is detailed in the data file (within the *FloatViz* structure). Briefly, they use the Lueker et al. (2000) $K_1^*$ and $K_2^*$, Perez and Fraga (1987) for K$_F$, Dickson (1990) for KSO$_4$, and Lee (2010) for total boron estimates. Both *[DSi]* and *[SRP]* are estimated from the measured nitrate concentration using stoichiometric ratios of 2.5 and 1/16, respectively. We applied the same method but substituted the Lueker et al. (2000) $K_1^*$ and $K_2^*$ constants with either the constants from this study or the Millero et al. (2002) constants. We were then able to compare surface-ocean (defined as the upper 10 m of the water column) *pCO$_2$* obtained using Lueker et al. (2000) or Millero et al. (2002) with *pCO$_2$* obtained using the constants from this study. The analytical uncertainties were set to $\sigma S_P = 0.005$ and $\sigma T = 0.005$ °C (Olsen et al., 2016), $\sigma[DSi] = 0.9$ µmol kg$^{-1}$ and $\sigma[SRP] = 0.5$ µmol kg$^{-1}$ (combination of uncertainty in nitrate concentration from Argo data, i.e., 0.5 µmol kg$^{-1}$ as given in Johnson et al. (2017) and a 30% uncertainty in stoichiometric ratios), $\sigma TA = 5.6$ µmol kg$^{-1}$ (Carter et al., 2018) and $\sigma pH = 0.005$ (Johnson et al., 2017). $\sigma pK_1^*$ and $\sigma pK_2^*$ were set to 0.011, respectively, when the constants from this study were used. For both Lueker et al. (2000) or Millero et al. (2002), they were set to the default values given by Orr et al. (2018), i.e., $\sigma pK_1^* = 0.0075$ and $\sigma pK_2^* = 0.015$. Uncertainties on the computed *pCO$_2$* were propagated using the Matlab version of the Orr et al. (2018) CO2SYS software with error propagation.

*pCO$_2$* values obtained with the constants derived from this study are clearly higher than the Lueker et al. (2000)-based values in the southernmost regions, where temperatures are lowest (Fig. 7a,b), with a maximum difference ($\Delta pCO_2 = pCO_2^{\text{Lueker}}$

$-pCO_2^{this\ study}$) of -55 ± 17 µatm when the surface ocean is near the freezing point. The uncertainty on $\Delta pCO_2$ ($\sigma\Delta pCO_2$, *gray*
*lines* in Fig. 7) is computed as the square root of the sum of the squares of $\sigma pCO_2^{Lueker}$ and $\sigma pCO_2^{this\ study}$. Given the large
uncertainties, the $pCO_2$ difference between values based on constants derived from this study and values based on Lueker et
al. (2000) is only statistically significant (i.e., $\Delta pCO_2 + \sigma\Delta pCO_2^{this\ study} < 0$) for temperatures below ~8 °C (Fig. 7a). $pCO_2$
values obtained using Millero et al. (2002) constants appear to be midway between $pCO_2$ values obtained using Lueker et al.
(2000) constants, and $pCO_2$ values based on the constants from this study (Fig. 7c,d).


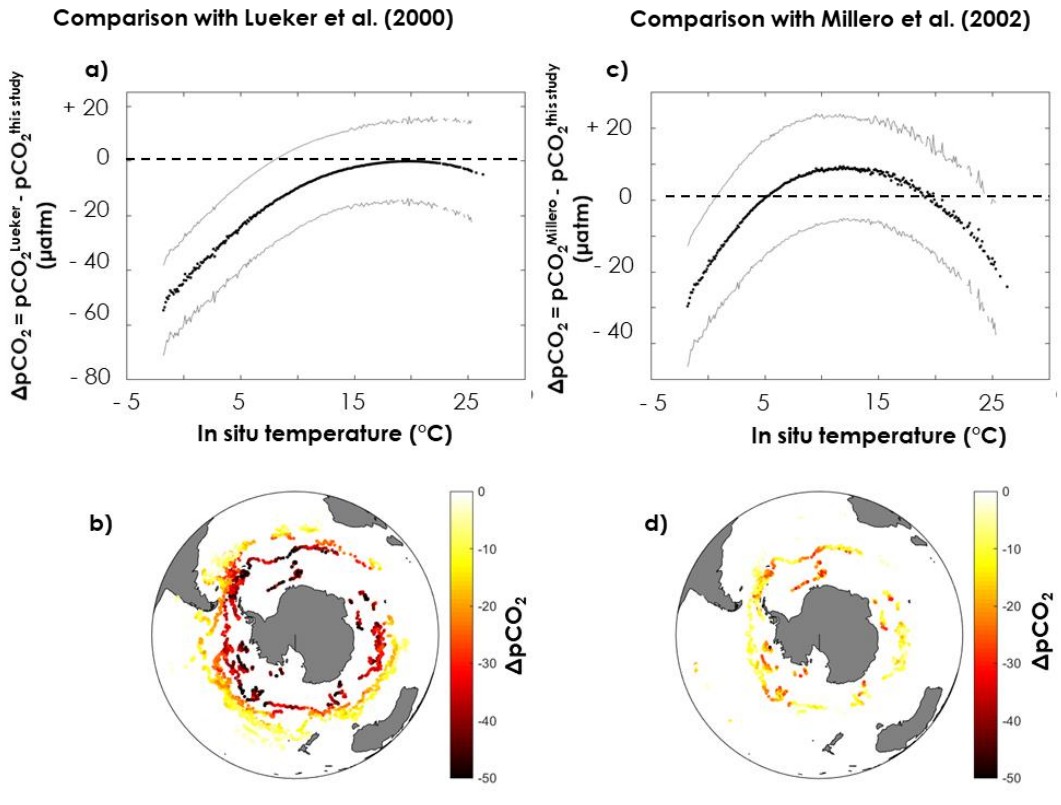

***Figure 7.*** *Difference in surface $pCO_2$ obtained from pH, temperature, practical salinity and dissolved oxygen*
*measured by the SOCCOM Argo array using (**left**) the Lueker et al. (2000) constants and the constants in Table*
*1, or (**right**) the Millero et al. (2002) constants and the constants in Table 1. Plots (**a,c**) represents the $pCO_2$*
*difference ($\Delta pCO_2$) as a function of in situ temperature. Solid black lines are the mean $\Delta pCO_2$ while solid gray*
*lines are plus or minus $\sigma\Delta pCO_2$ (square root of the sum of the squares of $\sigma pCO_2^{Lueker}$ and $\sigma pCO_2^{this\ study}$). Maps (**b,d**)*
*depict the spatial distribution of $\Delta pCO_2$ in the Southern Ocean, where each point corresponds to an Argo float*
*measurement.*


Recently the SOCCOM Argo array was used to re-evaluate the Southern Ocean carbon sink (Gray et al., 2018). Traditional ship-based observations indicate a strong $CO_2$ uptake in the entire Southern Ocean, but these observations are known to have a strong seasonal bias (Bakker et al. 2016), as well as a smaller spatial bias due to many areas being severely undersampled (Takahashi et al., 2012). Using $pCO_2$ calculated by the above method, Gray et al. (2018) showed that the

Southern Ocean $CO_2$ uptake is considerably smaller than previously estimated. In parallel, Bailey et al. (2018) showed that the $CO_2$ solubility constant from Weiss et al. (1974) used in the majority of studies, including this one, was underestimated in waters below 0 °C, which implies that surface $pCO_2$ is underestimated. In this study, using the new constants in Table 1, the computed Southern Ocean pCO2 is also higher than when computed using the constants of Lueker et al. (2000) or the constants of Miller et al. (2002), as shown in Fig. 7. The Southern Ocean is a net $CO_2$ sink because the $pCO_2$ in surface waters is in

average lower than the atmospheric $pCO_2$. If the surface-water pCO2 is revised upward, the resulting flux of $CO_2$ from the atmosphere to the surface waters becomes smaller. Thus, results from Gray et al. (2018), Bailey et al. (2018) and the present study all advocate for a weaker $CO_2$ sink in the Southern Ocean. The ocean $CO_2$ sink is immensely important, and currently estimated to remove ~25% of anthropogenic $CO_2$ emissions (Le Quéré et al., 2018). If the $CO_2$ uptake by the Southern Ocean is much smaller than previously estimated, there must be missing sinks elsewhere in the Earth System, be it in the oceanic or

terrestrial realm. This highlights the need for a better understanding of the dynamics of the ocean carbon sink, including its regional and temporal variability. To validate our results, the high uncertainties associated with stoichiometric constants (Orr et al., 2018), coupled to the low spatial and temporal resolution of measurements in high latitudes, need to be addressed. Whether in the laboratory or in the field, future work should focus on a better understanding of seawater carbonate chemistry in cold waters.

**4.4 Implications for calcium carbonate chemistry**

In seawater undersaturated with respect to calcite or aragonite, ($\Omega_{Ca}$ or $\Omega_{Ar} < 1$), the $CaCO_3$ phase of interest should dissolve if present. $\Omega_{Ca}$ depends on the $Ca^{2+}$ concentration in seawater (a function of salinity and therefore nearly invariant at depth), the solubility product of calcite (Mucci, 1983) and the $CO_3^{2-}$ concentration in seawater. Because of the latter, computed $\Omega_{Ca}$ values are impacted by the choice of carbonic acid dissociation constants. Note that, as reported in Orr et al. (2018), the

relative uncertainty on the solubility product of calcite is about 5%. Below, we test the implications of the $K_1^*$ and $K_2^*$ values from this study on predictions of calcite saturation state in seawater.

Naviaux et al. (2019) recently observed discrepancies between $\Omega_{Ca}$ computed with the *TA-DIC* pair and $\Omega_{Ca}$ computed with the *TA-pH* pair, that they attributed to the internal inconsistency of the carbonate system, i.e., the fact that measured *pH*

does not correspond to calculated *pH*. Instead, or in addition, the calcite saturation depth calculated by Naviaux et al. (2019) could be erroneously too shallow due to an overestimated $K_2^*$ and, consequently, overestimated seawater $[CO_3^{2-}]$ and $\Omega_{Ca}$.

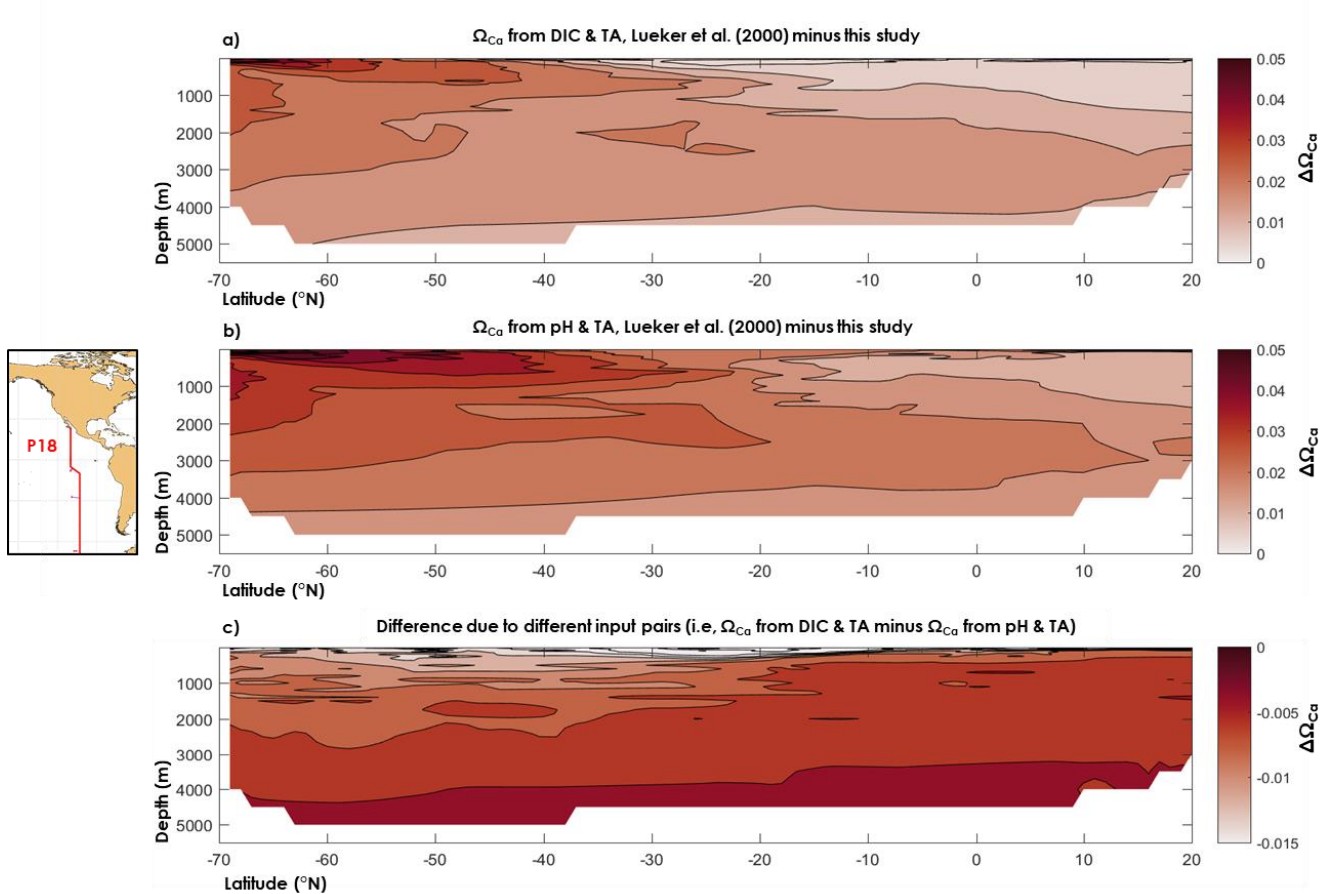

**Figure 8.** *Difference between a) the calcite saturation state ($\Omega_{CA}$) computed from TA and DIC using the dissociation constants of Lueker et al. (2000) and the constants derived from this study, b) $\Omega_{CA}$ computed from pH and TA using the dissociation constants of Lueker et al. (2000) and the constants derived from this study and c) $\Omega_{CA}$ computed from the DIC and TA pair and the pH and TA pair, and the dissociation constants derived from this study.*

Here, we used data from a cruise (33RO20071215, GLODAPv2 cruise #345) along the CLIVAR repeat section P18 that took place in 2007, following a latitudinal transect in the south-eastern Pacific Ocean, in which the carbonate chemistry variables *DIC*, *TA* and *pH* were measured (see Olsen et al. (2016) for details about the data). All calculations were carried out using the discrete data, but for purposes of visualisation (Fig. 8) we used a nearest neighbour interpolation (function *griddata* in Matlab). In Fig. 8, we compare $\Omega_{Ca}$ as computed using $K_1^*$ and $K_2^*$ from this study, with $\Omega_{Ca}$ based on Lueker et al. (2000) constants. We also compare $\Omega_{Ca}$ computed from *TA-DIC* with $\Omega_{Ca}$ computed from *TA-pH*. From Fig. 8, it can be seen that the $\Omega_{Ca}$ difference between two different carbonate system pairs (*TA-DIC*, *TA-pH*) is ~ 5 times smaller than the $\Omega_{Ca}$ difference that is due to the set of dissociation constants. Thus, the apparent dissolution observed by Naviaux et al. (2019) may be

explained by overestimated dissociation constants atop inconsistencies arising from the choice of carbonate variables used in

the calculations. We also note, based on Fig. 8, that $\Omega_{Ca}$ overestimation is largest in the southernmost part of the Pacific surface waters, where the temperature is the lowest. Nevertheless, the maximum calculated $\Omega_{Ca}$ differences, i.e., $\Delta\Omega_{Ca} = 0.06$ with *TA-DIC* and $\Delta\Omega_{Ca} = 0.07$ with *TA-pH*, is 2-3 times lower than the average combined uncertainty, i.e., $((\sigma\Omega_{Ca}^{\text{Lueker}})^2 + (\sigma\Omega_{Ca}^{\text{this study}})^2)^{0.5} = 0.20$ and $0.17$, respectively. These high uncertainties are attributed to the high measurement uncertainties that we use (those from Olsen et al., 2016, see section 2.4), the overall uncertainty on the dissociation constants from this study, the

uncertainty on Lueker et al. (2000) constants, but especially to the uncertainty in the calcite solubility product. As noted by Orr et al. (2018), the uncertainty on the calcite solubility product (~5%) causes the total uncertainty on $\Omega_{Ca}$ to be considerably larger than the uncertainty in seawater $[CO_3^{2-}]$. In this study, only the $K_1^*$ and $K_2^*$ constants have been re-evaluated, but the large overall uncertainties on calculated saturation states clearly indicate that more work is necessary to define the solubility products of calcium carbonate minerals in the ocean. While beyond the scope of the present study, the results presented here

show that proper assessments of present and future ocean acidification are highly sensitive to the present knowledge gaps regarding the thermodynamics of ocean carbon chemistry.

## 5 Conclusion

An iterative procedure allowed us to estimate the temperature dependence of the first and second carbonic acid

stoichiometric dissociation constants ($K_1^*$ and $K_2^*$, respectively) from a large dataset of high-quality oceanographic measurements. Both $K_1^*$ and $K_2^*$ were similar to the constants of Lueker et al. (2000) that are currently used by most of the oceanographic community, as recommended by Dickson (2007), but the $K_1^*$ and $K_2^*$ values were lower in cold seawater, below a temperature of ~8-9 °C. Consequently, at these temperatures, $pCO_2$ computed using the constants of Lueker et al. (2000) may be underestimated and $[CO_3^{2-}]$ overestimated, meaning that the cold oceans are more undersaturated with respect

to $CaCO_3$ minerals than expected. We also used a GLODAP sub-dataset to study the internal consistency of the carbonate system and found that the *DIC-pH* carbonate system pair provides the smallest overall uncertainty when computing seawater $pCO_2$. When calculating the saturation state of seawater with respect to calcite, the *TA-DIC* pair should be used to minimize the overall uncertainty when seawater is warmer than ~20 °C, whereas the *TA-pH* pair should be preferred below ~20 °C. These results are of critical importance for scientists contemplating studies of high-latitude marine carbonate chemistry and

underline that improved knowledge of what causes the $CO_2$ system inconsistencies in cold waters is key to improve our understanding of the marine carbon budget.

## Data availability

The R code and data files are available on Zenodo (https://doi.org/10.5281/zenodo.3725889).

## Author contribution

OS conceived the original idea, OS and MH designed the research, MH wrote the R code and SKL advised on the use of the GLODAPv2, SOCAT and Argo datasets. All authors contributed to manuscript writing, with OS taking the lead.

## Competing interests

The authors declare that they have no conflict of interest.

## Acknowledgements

We thank Dorothee C. E. Bakker and Mariana Ribas Ribas for fruitful discussions, Alfonso Mucci and Andrew Dickson for helpful comments and suggestions on an earlier version of the manuscript, the journal editor Dr. Mario Hoppema for handling the manuscript and two anonymous reviewers for their comments. We also thank Dr. Anna de Kluijver for her help in setting up the Monte Carlo simulation. We thank all who contributed to the creation of GLODAPv2 and SOCAT. OS acknowledges the Department of Earth and Planetary Sciences at McGill University for financial support during his residency in the graduate program and acknowledges funding from the Dutch Ministry of Education via the Netherlands Earth System Science Centre (NESSC). SKL acknowledges funding from the Research Council of Norway through (ICOS-Norway, 245927 and NorArgo2, 269753).

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
