# Peer review of "Current estimates of $K_1^*$ and $K_2^*$ appear inconsistent with measured $CO_2$ system parameters in cold oceanic regions"

_Ocean Science, 2020_

## Referee Comment (RC1) · Anonymous Referee #1 · 18 May 2020

General comments:

The manuscript is interesting, important and well written. The work is well described and well put into context.

Specific comments:

The short results section followed by a long discussion with new material and figures feels unusual, but after consideration I think it works very well. I also like the short and concise conclusion.

I do not have any major scientific comments or objections, I genuinely think the work is

well done, as far as I can tell.

Technical corrections:

line 47-48: "... activities or effective concentrations of the reaction products to the product of reactant activities... ". This is hard to follow, please rephrase.

line 98: "Barring" is an unusual choice of words. I personally like it, but suggest using something else for better accessibility of the text. I also suggest rewriting/inversing the sentence so that the main point comes first, and then the exception.

line 161: "WOCE flag of 2" please explain what this entails. There is an explanation much later in the MS, I suggest moving that to here.

line 272: "model fits" or "models fit" ?

line 288: Why the small changes in a2 in the table 1 ? Is it a typo?

line 298-303: I think this introduction-like text should come before lines 293-298. Possibly also before table 1.

line 376: Do the measurements face an issue? I think this is the wrong forum for that formulation. Do you mean to write that users of the data may face this issue?

line 409: Here is the explanation of the quality flag, please move it to earlier in the text.

line 488-494: A bit hard to follow. Especially "Gray et al. (2018) showed that the Southern Ocean CO2 uptake is considerably smaller than previously estimate" and then "Using the new constants in Table 1, Southern Ocean pCO2 is significantly higher than the values used by Gray et al. (2018), meaning that the air-sea CO2 fluxes are much smaller, in agreement with the conclusions of Bailey et al. (2018)".

Figure 8: It would be much better if the a, b and c panels could be placed directly below one another and be of equal size.

---

## Referee Comment (RC2) · Anonymous Referee #2 · 9 Jun 2020

General Comments: This clearly-written article is based on a sound analysis of GLO-DAPv2 and SOCAT data that reveals large discrepancies between the temperature and salinity dependence of published carbonic acid stoichiometric dissociation constants and relationships estimated from available data. These discrepancies are particularly extreme at cold temperatures and thus have enormous implications for understanding of carbon cycling and ocean acidification in high latitude ocean regions. This study further suggests that the uncertainty for carbonic acid dissociation constants at cold temperatures is more substantial than previous studies indicate and that the potential for bias in estimation of pCO2 from other carbonate system parameters is not well constrained. The factors that contribute to the inconsistencies between studies are

beyond the scope of the study and highlight a need for future studies based on contemporaneous in situ measurements of all four carbonate system parameters as well as laboratory studies.

---

## Author Comment (AC1) · 9 Jun 2020

Dear Reviewer #1

We are grateful for the time you have committed to providing feedback to this manuscript. We hope that our edits and responses will appropriately address all the issues that you raised.

**Reply to your comments of May 18 (os-2020-19-RC1.pdf)**

**General comments: The manuscript is interesting, important and well written. The work is well described and well put into context.**

**Specific comments: The short results section followed by a long discussion with new material and figures feels unusual, but after consideration I think it works very well. I also like the short and concise conclusion. I do not have any major scientific comments or objections, I genuinely think the work is well done, as far as I can tell.**

Thank you for these kind words!

**Technical corrections:**

**line 47-48: "... activities or effective concentrations of the reaction products to the product of reactant activities... ". This is hard to follow, please rephrase.**

The following sentence

"At equilibrium, the ratio of the activities or effective concentrations of the reaction products to the product of reactant activities that appear in Eqs. (2) and (3) yields a constant that describes the thermodynamic equilibrium of these reactions."

will be rewritten into:

"Each of these reversible reactions is associated with a thermodynamic equilibrium constant, a number that expresses the relationship between the activities of products and reactants present at equilibrium at a given temperature and pressure."

**line 98: "Barring" is an unusual choice of words. I personally like it, but suggest using something else for better accessibility of the text. I also suggest rewriting/inversing the sentence so that the main point comes first, and then the exception.**

We will replace "barring" by "except under" and restructure the sentence to: "Only two of the measurable variables are required to characterize the whole carbonate system, except under conditions where substantial impact of dissolved organic carbon on TA is expected (i.e. significant organic alkalinity)."

**line 161: "WOCE flag of 2" please explain what this entails. There is an explanation much later in the MS, I suggest moving that to here.**

We will add the following just after the mentioned sentence: "WOCE (World Ocean Circulation Experiment) flags are associated to each GLODAPv2 variable during quality control. Data associated with a flag of 2 were assessed as "acceptable" by quality controllers of the original dataset."

**line 272: "model fits" or "models fit" ?**

This was performed for both K1 and K2, hence will be changed to "model fits".

**line 288: Why the small changes in a2 in the table 1 ? Is it a typo?**

It is a typo and will be corrected, thank you!

**line 298-303: I think this introduction-like text should come before lines 293-298. Possibly also before table 1.**

We agree. We will place the introduction-like text before lines 293-298 and place the whole paragraph before Table 1.

**line 376: Do the measurements face an issue? I think this is the wrong form for that formulation. Do you mean to write that users of the data may face this issue?**

We will change this sentence to "Another issue in GLODAPv2 carbonate system measurements may be the fact that some seawater samples contain measurable amounts of organic bases"

**line 409: Here is the explanation of the quality flag, please move it to earlier in the text.**

We will do this in the revised version.

**line 488-494: A bit hard to follow. Especially "Gray et al. (2018) showed that the Southern Ocean CO2 uptake is considerably smaller than previously estimate" and then "Using the new constants in Table 1, Southern Ocean pCO2 is significantly higher than the values used by Gray et al. (2018), meaning that the air-sea CO2 fluxes are much smaller, in agreement with the conclusions of Bailey et al. (2018)".**

We agree with the reviewer. The point we wanted to make is that the Gray et al. (2018) study, the Bailey et al. (2018) study and our results all go in the same direction of a weaker than previously thought $CO_2$ sink in the Southern Ocean. We have rewritten the paragraph, which now reads as:

"Recently the SOCCOM Argo array was used to re-evaluate the Southern Ocean carbon sink (Gray et al., 2018). Traditional ship-based observations indicate a strong $CO_2$ uptake in the entire Southern Ocean, but these observations are known to have a strong seasonal bias (Bakker et al. 2016), as well as a smaller spatial bias due to many areas being severely undersampled (Takahashi et al., 2012). Using $pCO_2$ calculated by the above method, Gray et al. (2018) showed that the Southern Ocean $CO_2$ uptake is considerably smaller than previously estimated. In parallel, Bailey et al. (2018) showed that the $CO_2$ solubility constant from Weiss et al. (1974) used in the majority of studies, including this one, was underestimated in waters below 0 °C, which implies that surface $pCO_2$ is underestimated. In this study, using the new constants in Table 1, the computed Southern Ocean $pCO_2$ is also higher than when computed using the constants of Lueker et al. (2000) or the constants of Miller et al. (2002), as shown in Fig. 7. The Southern Ocean is a net $CO_2$ sink because the $pCO_2$ in surface waters is in average lower than the atmospheric $pCO_2$. If the surface-water $pCO_2$ is revised upward, the resulting

flux of $CO_2$ from the atmosphere to the surface waters is drawn downward. Thus, results from Gray et al. (2018), Bailey et al. (2018) and the present study all advocate for a weaker $CO_2$ sink in the Southern Ocean. The ocean $CO_2$ sink is immensely important, and currently estimated to remove ~25% of anthropogenic $CO_2$ emissions (Le Quéré et al., 2018). If the $CO_2$ uptake by the Southern Ocean is much smaller than previously estimated, there must be missing sinks elsewhere in the Earth System, be it in the oceanic or terrestrial realm. This highlights the need for a better understanding of the dynamics of the ocean carbon sink, including its regional and temporal variability. To validate our results, the high uncertainties associated with stoichiometric constants (Orr et al., 2018), coupled to the low spatial and temporal resolution of measurements in high latitudes, need to be addressed. Whether in the laboratory or in the field, future work should focus on a better understanding of seawater carbonate chemistry in cold waters."

**Figure 8: It would be much better if the a, b and c panels could be placed directly below one another and be of equal size**

We will restructure the figure to have panels of equal size.

We look forward to hearing from you regarding this submission and would be glad to respond to any further questions and comments that you may have. Once again, we thank you for your time!

Sincerely,

Olivier Sulpis, Siv K. Lauvset and Mathilde Hagens

June 9, 2020

---

## Author Comment (AC2) · 9 Jun 2020

Dear Reviewer #2

**Reply to your comments of June 9 (os-2020-19-RC2.pdf)**

**General comments: This clearly-written article is based on a sound analysis of GLODAPv2 and SOCAT data that reveals large discrepancies between the temperature and salinity dependence of published carbonic acid stoichiometric dissociation constants and relationships estimated from available data. These discrepancies are particularly extreme at cold temperatures and thus have enormous implications for understanding of carbon cycling and ocean acidification in high latitude ocean regions. This study further suggests that the uncertainty for carbonic acid dissociation constants at cold temperatures is more substantial than previous studies indicate and that the potential for bias in estimation of pCO2 from other carbonate system parameters is not well constrained. The factors that contribute to the inconsistencies between studies are beyond the scope of the study and highlight a need for future studies based on contemporaneous in situ measurements of all four carbonate system parameters as well as laboratory studies.**

We thank you for your time and for this review, and we agree that our work indicates the need for further measurements to reduce the uncertainty in the carbonic acid stoichiometric dissociation constants. We would be glad to respond to any further questions and comments that you may have.

Sincerely,

Olivier Sulpis, Siv K. Lauvset and Mathilde Hagens

June 9, 2020